# Research Progress on Applying Intelligent Sensors in Sports Science

**DOI:** 10.3390/s24227338

**Published:** 2024-11-17

**Authors:** Jingjing Zhao, Yulong Yang, Leng Bo, Jiantao Qi, Yongqiang Zhu

**Affiliations:** 1Physical Education Teaching Department, China University of Petroleum (East China), Qingdao 266580, China; zhaojj@upc.edu.cn; 2College of New Energy, China University of Petroleum (East China), Qingdao 266580, China; 2115010419@s.upc.edu.cn (Y.Y.); yqzhu2024@163.com (Y.Z.); 3College of Education, Beijing Sports University, Beijing 100091, China; lbihh@sina.com

**Keywords:** sensors, motion monitoring, sports, intelligent identification

## Abstract

Smart sensors represent a significant advancement in modern sports science, and their effective use enhances the ability to monitor and analyze athlete performance in real time. The integration of these sensors has enhanced the accuracy of data collection related to physical activity, biomechanics, and physiological responses, thus providing valuable insights for performance optimization, injury prevention, and rehabilitation. This paper provides an overview of the research progress in the application of smart sensors in the field of sports science; highlights the current advances, challenges, and future directions in the deployment of smart sensor technologies; and anticipates their transformative impact on sports science and athlete development.

## 1. Introduction

Sport, or physical activity, has a positive impact on quality of life. According to statistics, about 50% of Europeans are physically active at least once a week from the age of 15 [1]. Motion analysis involves the study of the movement of an object or organism through measurement and analysis. It includes the acquisition of kinetic and kinematic information, as well as the processing and interpretation of these data. Quantifying an athlete’s movements and positions during a sporting event is essential, which plays a key role in analyzing the state of the sport, identifying trends, predicting outcomes, and enabling coaches to assess athlete performance and physical condition. There are many parameters and conditions involved in complex sports, ranging from environmental conditions at the venue and the sports equipment to the movement and physical state of the athletes. To accurately capture movement information, various physical quantities must be monitored, including velocity, position, acceleration, and force load conditions [2].

The integration of smart sensors in sports science has been an area of growing interest in recent years and is now widely accepted and applied in various sports and fitness activities, from recreational to elite levels, and across both individual and team sports, including among non-disabled and disabled athletes [3]. These sensors, as key devices for information acquisition, together with communication and computer technologies, constitute the three technologies of modern information systems and are indispensable for information acquisition in various types of devices.

In addition, the rapid development of smart technologies has significantly advanced the field of wearable electronics, enabling enhanced intelligence in the acquisition, processing, analysis, and transmission of sensory data. Traditionally, signal analysis involved the manual extraction of basic features from sensory signals. However, smart technologies now empower wearable sensors to detect more complex and diverse signals, as well as to automatically extract key features that represent the underlying relationships within the data. By employing appropriate learning models tailored to specific sensing applications, these sensors can yield more comprehensive insights, thereby driving a transformative evolution in motion sensor technologies [4], for example, wearable IMU sensor systems and EMG sensors for recording and analyzing the electrical signals generated by muscle activity [5]. The combination of body-attached sensor devices and sensors integrated into sports equipment, along with appropriate sensor fusion algorithms, can aid in developing better sports equipment to accelerate learning processes and ultimately improve skill levels. Ishida et al. [6] emphasized the importance of sensor technologies and statistical analysis methods in formalizing sports science, particularly in identifying the key factors that contribute to successful skills. Therefore, the advantages of sensor technology in enhancing athletic performance and improving sport management systems are significant.

However, the sensor data acquisition process may face challenges, such as poor skin fit, humidity variations (e.g., sweat), and external environmental disturbances (e.g., temperature, magnetic fields), which can affect the accuracy and privacy of the device usage [7]. Addressing the issues faced by motion sensors in data acquisition, this paper highlights the principles and types of wearable sensors, summarizes the challenges of smart motion sensors in service, and explores future developments and applications of these sensing devices in sports science.

## 2. Materials and Methods

This systematic review was prepared according to the guidelines defined in Preferred Reporting Items for Systematic reviews and Meta-Analyses (PRISMA). It was conducted in the relevant databases, including Web of Science, PubMed, and IEEE Xplore, to identify works from 1 January 2010 until 31 June 2024 related to the application of intelligent sensors in sports science. The search utilized the following terms: TS = (intelligent sensors OR smart sensors OR wearable sensors) AND TS = (sports science OR sports training OR sports performance OR sports biomechanics OR sports medicine) AND TS = (application OR utilization OR implementation).

Studies were excluded if they fulfilled any of the subsequent criteria: firstly, those that were not related to sports or merely mentioned sensors in a general sense without specific application in the field of sports; secondly, reviews, commentaries, or papers that did not showcase original research regarding the utilization of intelligent sensors in sports; and lastly, studies that failed to offer explicit details about the sensor technology or its application within the sports context.

The relevant data were extracted from the papers that satisfied the inclusion and exclusion criteria. Specifically, they include the following aspects: (1) the design and methodology of the study, encompassing the type of sports activity, the number and characteristics of the participants, and the experimental setup; (2) the details of the intelligent sensor system, such as the type and number of sensors, the measured parameters, and the data processing algorithms; (3) the outcomes and results of the study, for example, the influence of the sensor system on sports performance, injury prevention, or training optimization.

Studies were classified based on the type of wearable sensors and the specific application area of the sensors (e.g., performance monitoring, injury detection, training feedback). Each study was reviewed by two independent researchers to ensure the accuracy of the data extraction.

## 3. Results and Discussion

### 3.1. Type of Wearable Sensors

Wearable sensors are increasingly being researched in the field of sports science, where wearable technology monitors the internal and external workloads of athletes and performs kinematic analysis. Recent advancements include the development of smart wearable sensors with high sensitivity, stretchability, light weight, and self-powered capabilities. These sensors can provide athletes with real-time feedback in their respective sports, thus helping them perform their roles more effectively. In recent years, there has been a gradual increase in research on wearable sensors, as shown in Figure 1. The implementation of multiple wearable sensors and data fusion techniques provides the potential for improved activity detection accuracy compared to single-sensor systems [8].

#### 3.1.1. Flexible Sensor

Flexible wearable sensors, encompassing a plethora of pressure sensor types—namely piezoresistive, capacitive, piezoelectric, and triboelectric—have garnered significant interest due to their expansive utility in health monitoring, human–computer interaction, artificial intelligence, and the Internet of Things. Among these four types, piezoresistive sensors have garnered significant attention due to their simple fabrication, straightforward readout mechanisms, low power consumption, and convenient signal acquisition. In recent years, graphene-based piezoresistive pressure sensors have emerged as a research hotspot, owing to their notable advantages, such as high flexibility, ultra-lightweight properties, and excellent mechanical strength. However, while the high elasticity and tensile properties of these sensors benefit from the use of organic materials, they also raise certain environmental concerns. Additionally, when these sensors are adhered to clothing or skin for detection, noise signals from the external environment can easily interfere with or even mask the desired signals. This challenge is not unique to graphene-based piezoresistive sensors, and their sensitivity to target signals can be enhanced by increasing the microstructure of the sensor [9]. Capacitive flexible pressure sensors are distinguished by their unique properties, including high energy efficiency, simple device structure, low manufacturing cost, and other benefits, making them essential for the development of high-sensitivity pressure sensors. However, these sensors face challenges such as insufficient detection limits, stability issues, and slower response times. Moreover, there is an increasing trend to use natural substances as templates for creating dielectric layer/electrode microstructures, replacing silicon-based templates fabricated via photolithography or electron beam lithography. Nevertheless, a trade-off between performance characteristics (e.g., sensitivity, stability, and cost) is often observed. Therefore, developing a cost-effective, high-performance flexible capacitive pressure sensor remains a key goal for future research [10]. Piezoelectric materials have been widely utilized in energy-harvesting applications for self-powered sensors and bio-integrated devices, owing to their ability to generate charges in response to mechanical deformations via the direct piezoelectric effect. These materials are characterized by high sensitivity, high power density, durability, scalability, mechanical stability, simple design, and ease of operation. However, the energy conversion efficiency of these materials needs further improvement [11]. Friction electric sensors are recognized for their high, stable output voltage, fast response recovery time, wide sensing range, good stability, and ultralow power consumption. One of the notable features of this type of sensor is its ability to provide a stable power supply from a constant voltage source [12]. However, conventional triboelectric nanogenerators (TENGs) face challenges such as uneven micro-level undulations when the two friction layer materials come into contact, which limits electron exchange and affects current density and power output. Additionally, their production process often involves high-end materials and precise manufacturing techniques, resulting in higher production costs.

In recent years, the fabrication of flexible strain sensors that exhibit both high sensitivity and a broad sensing range continues to present a formidable challenge. There has been a surge in the development of highly flexible and wearable stress sensors, leveraging diverse materials, structures, and transduction mechanisms. The morphological refinement of nano- and micro-scale sensing materials has proven to be a critical factor in achieving superior sensor performance, thereby driving advancements in this domain [13]. Takahiro et al. [14] developed a smart ping-pong paddle based on PZT/Si composites, with a total sensor thickness of about 50.1 μm, making it more flexible than the PZT/PVDF sensors. The composite film can be easily integrated into a flexible circuit board. By applying lateral and longitudinal forces to the ball relative to the paddle, a correlation between the output response voltage and both time and force can be observed, allowing for detailed identification of the position and manner of the stroke. Additionally, this technology can be used to recognize pitching stances in shot put and basketball. Figure 2 shows a 5 × 5 (20 mm pitch) ultrathin piezoelectric strain sensor array integrated on a PI flexible printed circuit (FPC). FPCs with sensor arrays can be easily mounted between wooden paddles and rubber due to their total thickness of less than 100 μm.

Yu et al. [15] prepared a fabric strain sensor with high sensitivity and wide sensing range by layer-by-layer self-assembly of poly (vinyl alcohol) and MXene layers using polyester elastic fabrics with a warp-knit weft-induced structure. Babu et al. [16] conducted a literature survey on the use of nanomaterials for the fabrication of stretchable strain sensors with special reference to healthcare, sports, and recreation fields. Ren et al. [17] also highlighted the application of low-cost, high-sensitivity, wide-detection-range flexible PZT sensors in the field of sports training.

#### 3.1.2. Electrochemical Sensors

Wearable electrochemical sensors utilize electrochemical reactions to detect the presence and concentration of specific substances. They typically include electrodes and electrolytes, and respond to target analytes by measuring changes in current, potential, or electrochemical impedance. These sensors are characterized by good flexibility, miniaturization, portability, excellent biocompatibility, and low cost of detection. They provide accurate, safe, and real-time non-invasive monitoring of physiological signals in sweat, which can help in understanding various physiological indicators of athletes during training, predicting sports risks, preventing sports injuries, and providing scientific guidance for sports training. This technology has significant potential for application in sports monitoring. Liu et al. [18] designed and synthesized sensors with enhanced performance using the unique properties of ZnO/TiO_2_ nanocomposites. The sensor has high sensitivity and selectivity for methyltestosterone, making it superior to traditional detection techniques and a powerful tool for anti-doping control.

Commonly used substrate materials for wearable electrochemical sensors include polymers, textiles, paper-based materials, hydrogels, and rubber polymers [19]. Polymer materials usually refer to plastic-based polymers, such as polyterephthalic acid (PET), polyimide (PI), and polyester (PE), as shown in Figure 3A. These materials are malleable, lightweight, chemically stable, and low-cost, making them widely used as flexible substrates and encapsulation materials for wearable electronics [20]. Textile materials include cotton, wool, silk, nylon, and polyester, as shown in Figure 3B, which are comfortable, lightweight, pliable, biocompatible, and easily absorbent. Paper-based materials have advantages, such as being low-cost, biodegradable, and renewable, as shown in Figure 3C, and are widely used in wearable electrochemical sensors. Hydrogel materials are mostly prepared from biopolymers such as cellulose and chitosan, providing good biocompatibility for wearable sensors, as shown in Figure 3D. Rubber, being a highly elastic polymer material like polydimethylsiloxane and co - polyester, exhibits excellent stretchability and elasticity. Consequently, sweat sensors based on it are more proximate to human skin and are highly suitable for sweat monitoring in sports scenarios, as shown in Figure 3E. These materials play various roles, such as storing electrolytes, increasing sensor flexibility, and sampling biofluids, and they have a 3D mesh structure that can swell rapidly in water without dissolving.

Su et al. [21] successfully detected tartrazine in sports drinks using a Co_3_O_4_-modified disposable electrochemical sensor with molecular imprinting, demonstrating good selectivity and sensitivity, with a detection limit of 33 nM. Su et al. [22] summarized advances in the electrochemical detection of illicit drugs in sports, including stimulants, sedatives, anesthetics, diuretics, anabolic hormones like nandrolone, and masking agents such as theophylline, contributing to the development of novel electrochemical sensors for drug analysis. Notwithstanding commendable progress in the realms of flexible materials and biosensing innovations, the maturation of wearable sweat-sensing technology has regrettably languished at the preliminary stages of experimentation and laboratory testing. This impasse primarily stems from multifaceted challenges, which include individual-to-individual variations, inconsistencies in sampling techniques, the precise designation of sweat-gathering locations, along with a deficient comprehension of the intricate physiological reactions manifest in perspiration. Furthermore, the paucity of exhaustive research elucidating the nexus between sweat and blood constituents exacerbates the quandary, hindering the progression toward viable, real-world applications [23], and biosensors, big data, advanced artificial intelligence and machine learning algorithms go hand in hand to provide a complete solution for wearable applications [24].

#### 3.1.3. Inertial Sensors

Inertial sensors are increasingly used to monitor human motion and biomechanics and to relate these data to daily life tasks, due to their ability to detect small changes in linear and radial inertia. These sensors have been successfully applied in various research projects on sport-related monitoring, and their positive impact is expanding into other areas such as education, business, and services [25]. Camomilla et al. [26] discussed the superiority of miniature magnetic inertial sensors for measuring athletic performance during training or competitions, using magnetic inertial technology to enhance injury prevention and training specificity, thereby prolonging and enhancing athletes’ careers. Hakim et al. [27] used an MPU6050 sensor, which includes an accelerometer and a gyroscope, in an IMU single chip. The gyroscope was set to measure up to 1000 degrees per second. Since the gyroscope measures the angular velocity of the motion, the proposed device has a significant data change in the gyroscope *y*-axis. If the foot moves forward, the sensor rotates on the *y*-axis. The sampling rate data are 33 Hz, so the prototype needs to store 33 data in 1 s. An Xbee transmitter mounted on the prototype sends the data to an Xbee receiver connected to a computer. After creating an on-purpose application to be used by this prototype for data retrieval, the obtained data transmitted wirelessly. The acquired data were saved into a file in Excel format.

As shown in the data presented in Figure 4a, a distinct pattern emerges during the step sequence. The intervals from point 1 to point 3, point 3 to point 5, and point 5 to point 7 exhibit two-step waveforms. The movement from point 1 to point 3 corresponds to the motion of the foot, as illustrated in Figure 4b. Specifically, the change in the waveform from point 1 to point 2 in Figure 4a reflects the movement of the foot during action 1 to action 2, as the device is attached to the right foot pedal. During this phase, the angular velocity is significantly measured by the device. In contrast, the change in angular velocity between point 2 and point 3 is less pronounced, as shown in Figure 4b, when motion 3 and motion 4 occur. This is attributed to the inward movement of the left foot. This pattern can be utilized to determine the frequency value of a two-step cycle based on the time domain. As shown in Figure 4c, the wave from point a to point b represents a 2-foot stride. Point a to point b consists of 65 data points. With a sampling rate of 33 H, the period to produce 65 data points is 1.95 s, corresponding to two steps. The analysis of fluctuation points during the step yielded a footstep frequency of 0.513 H. Furthermore, from the obtained time period, the linear velocity was calculated to be 0.39 m/s or 1.404 km/h, thus completing the study of human gait monitoring, particularly footfall frequency and linear velocity. The relative error was utilized to compare the linear velocity of the treadmill with the measured values, and the Fast Fourier Transform (FFT) was used to validate the results of the time-domain data and accurately measure the exercise step frequency.

Ooi et al. [28] investigated the use of wearable inertial sensors and neural networks (NN) to recognize badminton strokes, developing an automated window segmentation method to identify stroke instances. A scaled conjugate gradient training algorithm with two hidden layers and 55 neurons per layer was identified as the best method for classifying badminton strokes, achieving an accuracy of 97.69%. Although inertial sensors are the most widely used for assessing athletes’ performance, there are also applications involving the combination of force sensors and electromyography in this field [29].

### 3.2. Challenge in Intelligent Sensor

The development of motion analysis technology should focus on improving performance in terms of accuracy, acquisition speed, and response time, while accurately reflecting detected data. Motion-sensing interference can primarily come from external factors such as corrosion and signal perturbation. When monitoring human movement, sweat secretion can significantly affect the performance of sensors in contact with the skin [30]. Additionally, wearable sensors in aquatic environments, such as swimming and diving, can be exposed to conditions that may cause short-circuiting, damage to sensing components, and lead to the decomposition, detachment, or oxidation of conductive materials [31,32]. Given these challenges, research has focused on developing motion sensors with corrosion resistance and high hydrophobicity. Zhou et al. [33] proposed a multifunctional superhydrophobic flexible sensor prepared using template methods and laser direct-writing technology. Micro-pillar array superhydrophobic L-CNT@PDMS sensors were prepared using a template-curing method and picosecond laser direct ablation. The L-CNT@PDMS sensors exhibit excellent corrosion resistance, superhydrophobicity, and anti-deicing properties while maintaining high tensile strength and electrical conductivity. Gao et al. [34] developed flexible, breathable, and corrosion-resistant wearable strain sensors with high tensile properties and sensitivity. These sensors can maintain their superhydrophobicity and electrical conductivity even after composite SCNC is subjected to cyclic stretching, abrasion, or harsh conditions. Figure 5a–f illustrate the morphological evolution of SiO_2_/graphene/PU composites during stretching and release. Stretching the nanofiber composites to generate cracks partially exposes the polymer nanofibers to the graphene shell. The crack morphology is marked by the red circles in this figure, and the size of these cracks ranges from a few nanometers to hundreds of nanometers, depending on the applied strain. Figure 5a shows the unstretched SCNC morphology, with SiO_2_ nanoparticles distributed on the surface of the wrinkled graphene shell. When the SCNC is stretched by 20% (Figure 5b), narrow microcracks appear, and a slight break in the conductive pathway causes a small increase in resistance. As the SCNC is further stretched to 50% (Figure 5c) and 100% (Figure 5d), the wrinkled graphene shell becomes relatively smooth, and more extensive microcracks appear, leading to a severe disconnection of the conductive network and a corresponding increase in resistance. Additionally, silica nanoparticles can act as “defects” in the graphene shell, making the conductive network more susceptible to strain, resulting in a higher resistive response compared to the graphene/PU-100% strain sensor. During the release process, the cracks narrow and blur as the strain is reduced from 100% to 50% (Figure 5e) and then to 20% (Figure 5f). When the SCNC is fully released (ε = 0) (Figure 5a), the resistance of the strain sensor returns to its initial value, demonstrating excellent reproducibility.

Ma et al. [35] developed a novel micro/nanofiber-coupled superhydrophobic conductive SNWTC with the ability to operate in extreme underwater environments, demonstrating good underwater sensing performance. This advancement provides an effective approach for creating wearable sensors suitable for marine applications, with significant implications for marine environment reconnaissance, resource development, and biological research. Ma et al. [36] prepared a superhydrophobic e-textile with a synergistically coupled dual conductive coating, where the first layer is a conductive Ti3C2TX (MXene) and the second layer consists of an electrically conductive composite with a layered papillary structure. The outer coating protects the inner MXene layer from oxidation and provides a superhydrophobic surface, further broadening the application range of the D-textile in wearable electronic devices, such as sweaty, rainy and even underwater conditions, which is valuable in all human activities performed in high-humidity environments.

Additional challenges include optimizing advanced functions of wearable devices—such as continuous data processing, sensor operations, and inter-device communication—which require high power and result in faster battery consumption. Therefore, enhancing energy efficiency in these operations remains an ongoing challenge.

At the material level, developing breathable, flexible, and stretchable materials (e.g., ultra-flexible wood 214) is a critical challenge to meet the stringent requirements of wearable applications, such as adapting to e-skins, smart patches, or textiles. The primary focus should be on enhancing the overall performance of the sensor, rather than excessively emphasizing individual attributes. Current research primarily concentrates on performance metrics such as stretchability and sensitivity, with significant advancements having been made in these areas. However, given that sensor signals can be easily amplified through specialized electronic devices, expanding the sensing range has become more critical than further improving sensitivity. Consequently, additional research efforts should be directed towards developing strain sensors that exhibit both a high gauge factor (GF) and a broad stretching range [37].

At the sustainable utilization level, developing self-powered wearable devices, including “green” power units (e.g., disposable solar panels or biofuel cells) or powerless options via near-field communication, is essential to ensure the continued use of wearable sensors [38]. Moreover, the sustainable and low-cost mass production of wearable sensors necessitates the use of transient and recyclable (even compostable) substrate materials [39]. Focus can be placed on the development of TENGs based on renewable, sustainable, and biodegradable materials (e.g., sodium alginate (SA), natural wood), the development of novel energy harvesting technologies for sustainable power supplies, and the realization of environmentally friendly sensor applications for sustainable self-powering [40,41].

At the data level, ensuring high data reliability is crucial. Interference from factors such as noise, temperature, humidity [42], dryness, and electromagnetism is always present during sensor use, and these issues can stem from both hardware and software problems. Additionally, user behavior during physical activities may introduce noise and artifacts, complicating the acquisition of high-quality wearable sensor data. Ensuring sufficient data volume and continuity is therefore essential. This issue can be mitigated through circuit optimization techniques, such as the use of filters and amplifiers, as well as advanced data-processing methods. To extract the true signal, an initial threshold can be established, followed by the application of a threshold evaluation algorithm to filter out output signals that fall below this set threshold. Additionally, various deformation patterns induced by the input stimulus can be detected and further decoupled using artificial neural networks.

### 3.3. Application of Intelligent Sensors

#### 3.3.1. Object-Oriented Motion Detection

With the increasing use of sensor technology in sports, motion-monitoring systems that identify and localize fast-moving objects play an important role in all types of competitions, both in competition and in daily training. Fan et al. [43] applied target detection and tracking technology to intelligent table tennis training scenarios by collecting players’ game videos, accurately identifying ball drop points, and analyzing drop zones. The system algorithms were tested multiple times within the project team’s intelligent training system, successfully identifying drop points and scoring zones with high accuracy and real-time performance. Vette et al. [44] outlined methods for estimating mechanical power output during periodic motion using wearable sensors, offering recommendations for mechanical power measurements across various sports. It is worth emphasizing that appropriate kinematics and forces can be measured using IMUs or strain gauges when power meters are either unavailable or impractical. The future prospects for mechanical power estimation using IMUs appear particularly promising, with potential for significant advancements through the integration of machine learning techniques. However, in specific cases, such as during wheelchair propulsion, force measurement via IMUs may be redundant. In such instances, mechanical power can be estimated by combining rolling resistance losses with kinetic energy estimation. Yang et al. [45] pioneered a methodology for the real-time surveillance of stress dispersion atop the foot, utilizing a triboelectric nanogenerator (TENG) integrated with an in-shoe sensor pad (ISSP) seamlessly embedded within the footwear’s upper lining. Capable of sensing and recognizing the static and motion comfort of a shoe, it also shows the health of the shoe and the concentration of pressure on the toes. By studying the static comfort degree, the link between the detected pressure and the comfort feeling is established. Then, the change in the comfort degree of shoes during various sports games can be sensed, which provides references for the intelligent manufacturing of shoes (Figure 6a). First, the characteristic signal outputs of four movements were measured: heel lift—off from the ground; forefoot off the ground (d_1_ = 2d_2_); foot tilted to the left and landed on the right side; and foot tilted to the right and landed on the left side. The signal output results are presented in Figure 6b, from which it can be observed that each output channel of the ISSP can generate different signals corresponding to the pressure changes at different parts of the foot as the foot movement varies. Figure 6c shows the visualization of ISSP data transmitted wirelessly during movement. This innovative ISSP offers a prospective avenue for scrutinizing foot kinematics and assessing comfort over extended periods, providing guidance for foot monitoring and customized shoe design.

Umek et al. innovatively harnessed an array of sensors to concurrently gauge the kinetic intricacies of skiers’ maneuvers alongside the reactive dynamics of the skis and the terrain they traverse. Data gleaned from these intricately integrated sensors serve a spectrum of objectives. This study applied several sets of sensing devices, such as force and bending sensors, inertial sensors, etc., in which IMUs were used to study the motion state of the human body, which shows the importance of IMUs in motion science monitoring. The study aids skiers in refining their skills, facilitating acceleration while ensuring proper technique. Additionally, it equips instructors with insights into identifying and rectifying flaws in skiing methods. Furthermore, it supports learners by outlining structured exercises tailored for instructional guidance, thereby enhancing the efficacy of ski tutelage [46,47].

Rennane et al. introduced a streamlined approach for accurately gauging the pressure exerted on a ball, employing a force-sensitive resistive sensor interfaced with a UHF Radio Frequency Identification (RFID) tag. This setup proved its efficacy through meticulous comparison with readings obtained from a calibrated electronic ball pressure gauge. Moreover, this technological advancement hints at the broader potential to surveil various forms of sporting equipment, including bats, during active gameplay, thereby ushering in new era of data-driven athletic performance analysis [48,49].

#### 3.3.2. Human-Oriented Monitoring

The scrutiny of athletic workload throughout training and competition has ignited fervent interest among both scientists and coaching professionals, transforming into a paramount focus in contemporary sports science. Adopting a comprehensive, multi-faceted strategy, experts relentlessly pursue optimal methodologies to amass and decipher performance data, catalyzing a surge in empirically rich and practical research. The domain has evolved with such velocity in recent times that it has given rise to a burgeoning sector dedicated to pioneering groundbreaking frameworks. These innovations enable precise quantification of the internal stresses and external pressures endured by athletes, thereby playing a pivotal role in safeguarding them against injuries and promoting robust health [50]. Hribernik et al. [51] proposed an advanced method for measuring human motion in agility tests by combining infrared optical gates and motion sensors. Figure 7 presents one of the numerous motion signals obtained in this study. For instance, at the commencement and conclusion of the test, the athlete stands upright and the field is approximately 90°. In this plane, it was feasible to measure the angle of approach at the start, as well as the time required to reach the central door, where an orientation change occurred. Substantial inclination changes were also noticed during the change of direction between the left and right gates. After the 8th second, a change in the backward stroke during this period was also observed. This method integrates precise time measurement, position detection, and motion tracking to capture previously inaccessible kinematic and spatiotemporal variables, improving the accuracy of agility assessments and providing additional insights for athletes and coaches. Human motion across three axes was measured using multiple motion sensors, synchronized with spatiotemporal parameters obtained from optical gates. Once determined, these measurements can be used for advanced agility assessment.

Biró et al. [52] analyzed the use of medical radar sensors and triaxial accelerometer data to predict physical activity KPIs in high-performance sports through machine learning. Flao et al. [53] conducted a systematic review of head impact research utilizing inertial sensors in sports, emphasizing the critical need to understand the factors contributing to head impact exposure for the development of effective prevention strategies. This review improves the understanding of sport-related head impacts, an area that is still evolving across many sports and requires technological advancements and standardized processes.

Liu et al. [54] identified and discussed the significance of revealing player performance and physical condition metrics, highlighting the rapid advancement of flexible and wearable motion sensor technologies that have transformed the way athletes’ movements and physiological states are captured. EMG, EEG, and temperature sensors have been used for in vivo and endurance exercise monitoring, as shown in Figure 8. The development of body-temperature sensors has focused on achieving wearability, high sensitivity, accuracy, portability, large array coverage, and real-time monitoring capabilities. Typically, RTDs, thermocouples, and thermistors are employed to convert temperature changes into corresponding electrical signals. These sensors can be ultrathin, with low heat capacity and fast response times. In contrast, myoelectric sensors use flexible dry electrodes to capture electromyograms. When combined with highly conductive nanomaterials such as metals, carbon, and polymers, these sensors provide a good fit to the skin and are effective in reducing noise interference. However, it is important to note that the complex and dynamic environment of the sports field can lead to significant deviations in sensor measurements due to temperature fluctuations, while mechanical shocks and external wear during movement may degrade device functionality. Therefore, integrating temperature sensors into various sensing systems should be considered to mitigate the adverse effects of these challenges through subsequent compensation algorithms.

#### 3.3.3. Cloud-Based Data Applications

The monitoring of human state parameters using sensing technology has numerous precedents and has been commercialized to some extent. Systematic, objective, and reliable performance monitoring and evaluation—through qualitative and quantitative analysis of performance variables—can strengthen the connection between research and training practices, particularly in elite sports. Wei et al. [55] proposed a scheduling strategy based on machine learning, combining reinforcement learning algorithms with deep reinforcement learning to set key factors for reinforcement learning and apply them to real-time motion images of athletes. This approach combines athletes’ motion characteristics to set actions and reward values. The algorithm is then used to allocate a reasonable data transmission path based on the real-time state, reducing network latency and providing a novel method for analyzing real-time sports images and body fat percentages of athletes, as shown in Figure 9.

Thomas et al. [56] applied and evaluated 12 estimation methods to address missing values in datasets from large-scale environmental monitoring, providing a tool to maximize the benefits of costly monitoring efforts. Zhao et al. [57] designed a data fusion method using isomorphic smart sensors to leverage smart mobile networks for identifying issues in early childhood physical education classes and detecting problems that hinder the development of early childhood physical education.

Alzahrani et al. [58] investigated the impact of wearable technologies, including inertial sensors, EMG sensors, and pressure sensors, on sports performance monitoring and intervention strategies in the field of physical therapy. Lyu et al. [59] proposed a novel model for physical activity recognition based on machine learning techniques. As shown in Figure 10, two convolutional components are used to process each input sample for recognizing patterns in the data through their convolutional filters. The first convolutional component extracts the primary data patterns, while the second convolutional component identifies deeper patterns. The objective is to effectively distinguish between physical and daily activities.

Building on this foundation, the integration of sports data monitoring with technologies such as the Internet of Things (IoT) and cloud computing, which enable automatic data analysis, will advance sports sensing and monitoring to new heights. The significance of big data acquisition and analysis in the advancement of smart sports cannot be overstated. Smart devices designed in array formats facilitate the collection of vast amounts of data for sports statistics and analysis. These devices are integrated into various sports equipment or environments based on specific needs, offering scientific guidance for sports activities, In the era of the Internet of Things (IoT), the acquisition and analysis of motion big data through widely distributed sensor networks play a crucial role in the development of intelligent motion systems. Conventional sensors typically rely on external power sources, which present limitations such as a finite lifespan and high maintenance costs. As a newly developed technology for mechanical energy harvesting and self-powered sensing, triboelectric nanogenerators (TENGs) offer significant potential to overcome these challenges. Figure 11 illustrates the constitution of the TENG-based selfpowered smart system. First, TENGs could be installed in sports facilities to record various mechanical trigger signals during sports activities. To monitor the physiological signals of the human body, TENGs can also be designed as wearable devices. Such an enormous number of TENG-based smart devices could be connected together into a network and widely applied in the sports domain. With the help of IoT, sports big data analytics and cloud computing technologies, sports training and competition will become smarter in the future. TENGs can function as self-powered sensors for detecting tactility, pressure, acceleration, or motion without requiring an external power supply. Furthermore, the enhancement of friction-based electric sensing signals and wireless sensing capabilities for TENG-based systems can be achieved through integration with signal processing and transmission modules. Therefore, the diverse applications of TENGs should be emphasized in the advancement of motion sensor intelligence [60].

## 4. Conclusions and Prospects

The proliferation of sensing devices plays a crucial role in digitizing motion analysis, with the degree of coupling between the sensing device, technological process, and user significantly influencing data accuracy. Factors such as environmental conditions, sports equipment, and the athlete’s movement and physical state can affect monitoring outcomes. This paper reviews the current state of research on wearable sensors and devices, focusing on flexible, electrochemical, and inertial principles. Although research on smart sensing in sports science is currently limited, it shows great promise. With the advancement of smart technology, sensors are expected to evolve towards higher precision, lower power consumption, and greater intelligence.

Additionally, sensors can wirelessly transmit sports data to external platforms, contributing to the creation of large-scale sports data repositories. Sensors primarily handle the front-end acquisition and collection of data for big data platforms. The collected sports data are processed using artificial intelligence and cloud computing technologies, which can enhance athletic performance, aid in developing scientific training plans and competition strategies, prevent injuries through automated data analysis, and create personalized databases for athletes to guide their daily training. This represents the future direction for sensor technology in sports. The integration of nanotechnology and new materials will also enable sensors to become more compact and sensitive, potentially allowing them to be embedded in clothing or skin, thereby achieving seamless, unobtrusive monitoring.

## Figures and Tables

**Figure 1 sensors-24-07338-f001:**
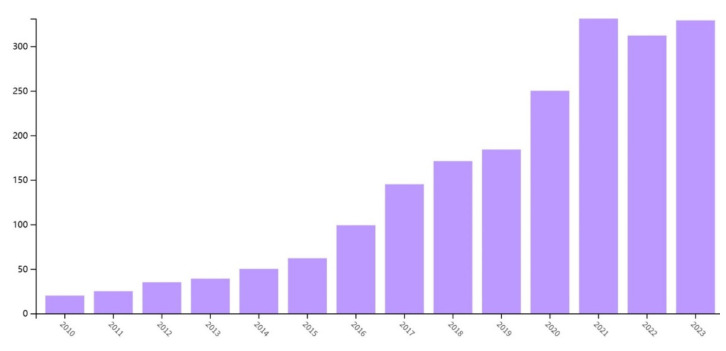
Annual publications of wearable sensors in sports science [data from Web of Science].

**Figure 2 sensors-24-07338-f002:**
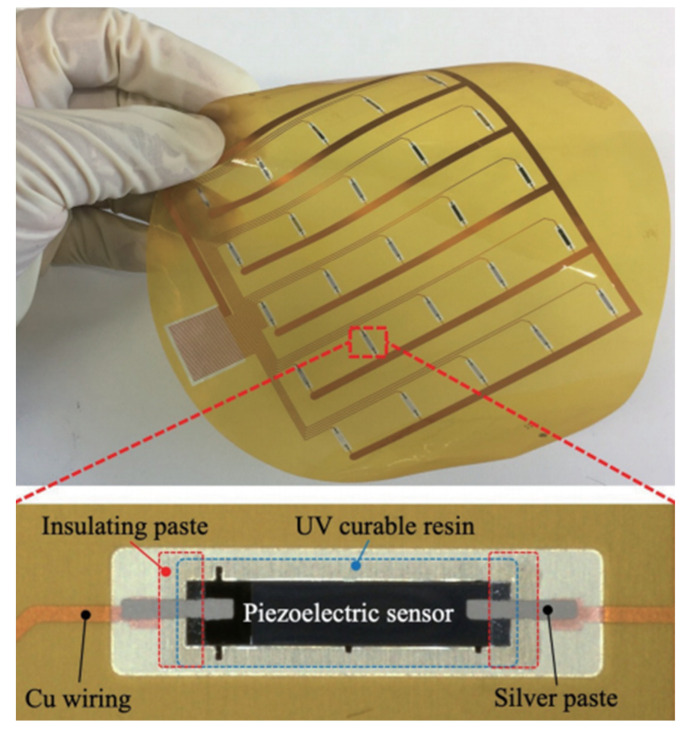
FPC with ultrathin piezoelectric sensor array [14].

**Figure 3 sensors-24-07338-f003:**
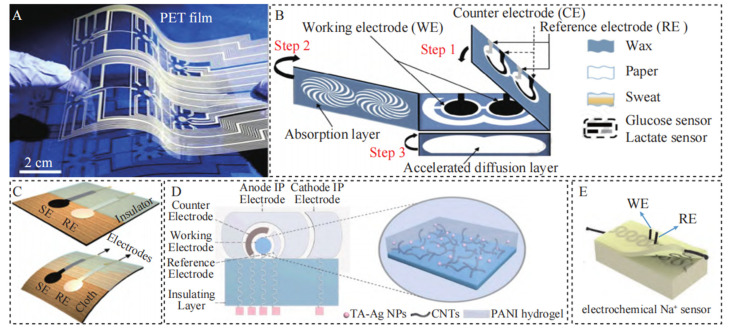
Different sensing substrates of wearable electrochemical sensors for sweat monitoring (**A**) Plastic; (**B**) Textile; (**C**) Paper; (**D**) Hydrogel; (**E**) Rubber [19].

**Figure 4 sensors-24-07338-f004:**
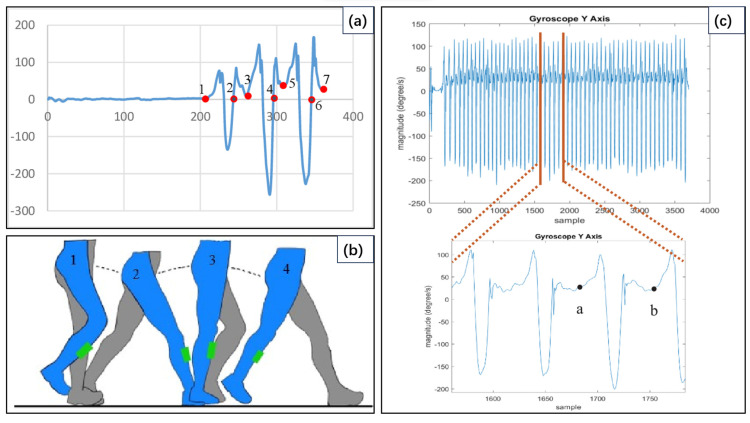
(**a**) Graph of the gyroscope sensor data on the *y*-axis for 6-foot steps; (**b**) Footsteps for one cycle; (**c**) Representation graph of the *y*-axis of gyroscope where point a is 1683 samples and point b is 1748 samples [27].

**Figure 5 sensors-24-07338-f005:**
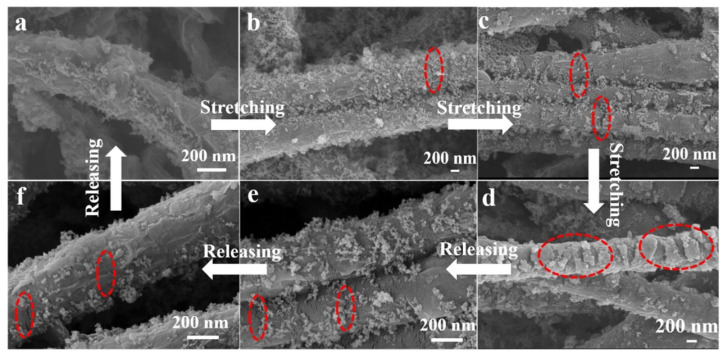
SEM images of the SCNC under different stretching and releasing states: (**a**) ε =10%. (**b**) ε = 20%. (**c**) ε = 50%. (**d**) ε = 100%. (**e**) ε = 50%, (**f**) ε = 20% [34].

**Figure 6 sensors-24-07338-f006:**
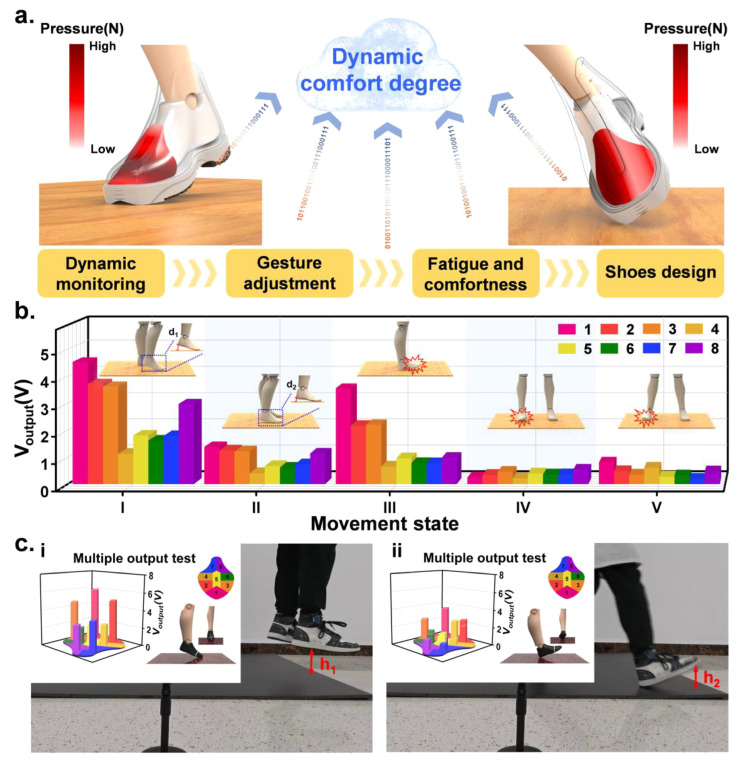
Application of the ISSP in dynamic responding with multiple information sources during the motions. (**a**) Schematic diagram of the ISSP for detecting foot movements, monitoring dynamic comfort degree of the shoe, and establishing the continuous movement model of instep and toes; (**b**) measurement of the different foot movements: (I, II) heel lifting d1 and d2 from the ground relatively, in which d1 = 2d2; (III) forefoot lifting from the ground; (IV, V) foot leaning to the left (right) with right-side (left-side) landing on the ground; (**c**) (i,ii) demonstration of foot movements’ wireless monitoring [45].

**Figure 7 sensors-24-07338-f007:**
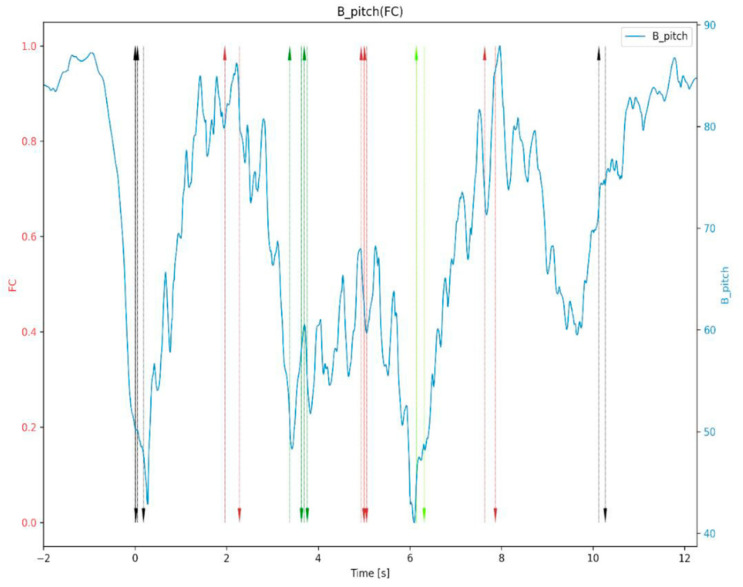
T-test graph containing optical gates data (vertical lines) and pitch signal from BNO055 (blue curve); at the beginning and at the end of the test, the athlete is standing upright, and the pitch is approximately 90° (the difference of a few degrees is because of the sensor mount at the lower back) [51].

**Figure 8 sensors-24-07338-f008:**
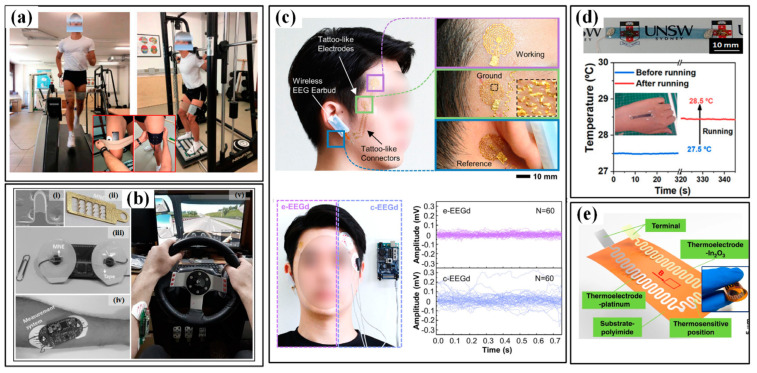
The EMG, EEG, and temperature sensors for sport monitoring: (**a**) the EMG sensor to monitor the strength and endurance exercises in vivo; (**b**) the microneedle array electrode−based wearable EMG system to detect driver drowsiness: (i) is the SEM photo of one single needle, (ii) is the photo of the microneedle array electrode, (iii) is the wearable EMG system; and (iv,v) are the system worn on forearm and driving; (**c**) the earbud-like wireless EEG device (up) show a good ability to decrease direct noise (down); (**d**) the wearable temperature sensor (up) and the measured small rise of skin temperature before and after a 5 min running exercise (down); (**e**) thin thermocouple’s film [54].

**Figure 9 sensors-24-07338-f009:**
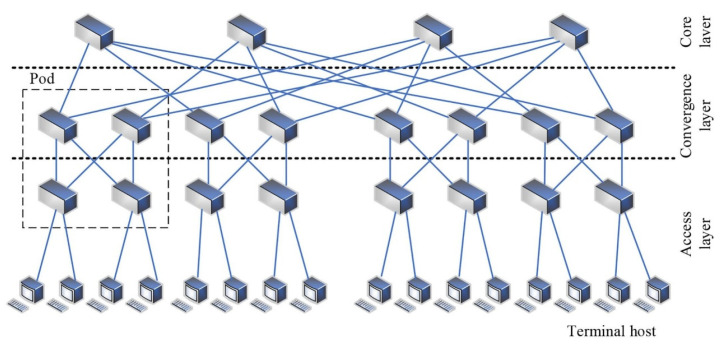
SDN data center network [55].

**Figure 10 sensors-24-07338-f010:**
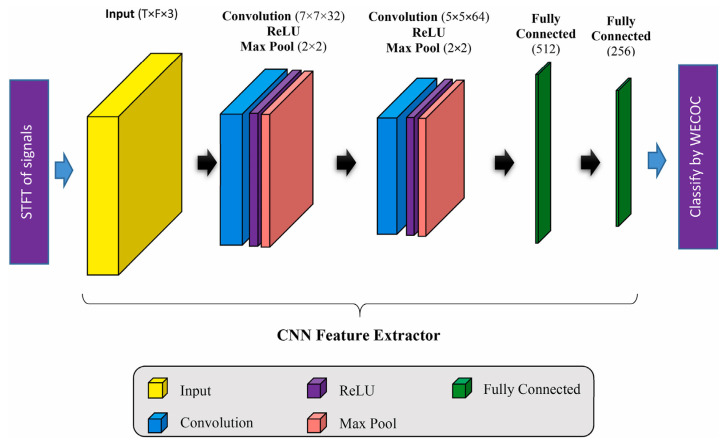
Structure of the proposed CNN model for extracting motion features [59].

**Figure 11 sensors-24-07338-f011:**
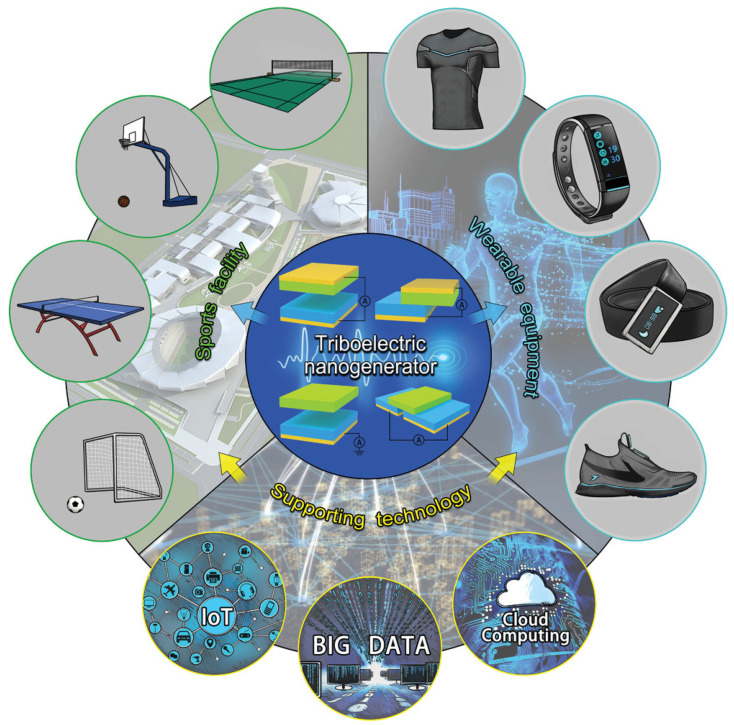
Schematic illustration of TENGs for intelligent sports based on the IoT, big data, and cloud computing technologies [60].

## Data Availability

No new data were created or analyzed in this study. Data sharing is not applicable to this article.

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
