# Peer review of "Research Progress on Applying Intelligent Sensors in Sports Science"

_sensors, 2024, doi:10.3390/s24227338_

Round 1
Reviewer 1 Report
Comments and Suggestions for Authors
This manuscript provides an overview of the research progress in the application of smart sensor in the field of sports science, highlights the current advances, challenges, and future directions in the development of smart sensor technologies, and anticipates their transformative impact on sports science and athlete development. However, there are some points needing to be clarified. The detailed points are as follows:
1. There needs to be attention paid to the author's name format to ensure consistency.
2. Please explain the meaning of points 1-7 in Figure 4. Besides, please explain the process of data extraction in detail, including how to deal with data inconsistencies and how to ensure the accuracy of data extraction.
3. It is recommended to check the resolution of figures to ensure they are clear.
4. The format of references is not standardized, and it is suggested to be revised and improved.
5. Please pay attention to the layout of the full text to make the layout of the article more beautiful.
6. This article conducts an extensive literature review on the application of intelligent sensors in sports science, covering various aspects such as motion monitoring and physiological signal detection. However, it is suggested that the authors could further highlight the advantages, limitations, and applicable scenarios of different sensor technologies in the review, so that readers can have a more comprehensive understanding of the applicability of various technologies.
Author Response
Please see in the attachment.

Reviewer 2 Report
Comments and Suggestions for Authors
The authors Jingjing Zhao et al summarized the advances in intelligent sensors applying on sports science, involving the current advances, challenges, and future directions in the deployment of smart sensor technologies. In overall, it is a relative comprehensive review paper to summarize the timely development of these intelligent sensors. Authors should resolve the following concerns before publication.
1. Many types of flexible wearable sensors have been proposed and developed for many years, such as y piezoresistive sensors, piezocapacitive sensors, piezoelectric sensors, and triboelectric sensors. Authors should compare the advantages and disadvantages of those sensors.
2. Authors should use the professional vocabulary. For example, friction electric should be triboelectric.
3. The title of this paper is “Research Progress of the Intelligent Sensors Applying on Sports Science. However, this manuscript mainly provides an overview of the wearable sensors. So, authors should point out that what are the different and the relationship between the intelligent sensors and wearable sensors.
4. In the part 3.2 of challenge in intelligent sensor, authors should give possible strategies, what should be done for future researchers to solve these problems.
5. What's the meaning of Figure 11? I haven’t found any meaningful discussion about this figure. Furthermore, it shows the TENGs for intelligent sports, rather than intelligent sensors or wearable sensors.
6. There are many errors in the manuscript. The names of the authors should apply the same style; On Page 10, Fig.6 shows should be “Fig. 9”; On Page 11, Figure 9 shows should be “Fig. 10”
7. The Figures can be further embellished. For example, Figure 4, 7.
8. Some typical work should be cited in the revised manuscript, such as doi.org/10.1038/s41467-019-13166-6”, doi.org/10.1016/j.cej.2023.142576 , doi.org/10.1016/j.cej.2023.143572.
Author Response
Please see in the attachment.

Round 2
Reviewer 1 Report
Comments and Suggestions for Authors
This manuscript can be accepted in present form.